# *ReliK*: A Reliability Measure for Knowledge Graph Embeddings

## ABSTRACT

*Can we assess a priori how well a knowledge graph embedding will perform on a specific downstream task and in a specific part of the knowledge graph?* Knowledge graph embeddings (KGEs) represent entities (e.g., "da Vinci," "Mona Lisa") and relationships (e.g., "painted") of a knowledge graph (KG) as vectors. KGEs are generated by optimizing an *embedding score*, which assesses whether a triple (e.g., "da Vinci," "painted," "Mona Lisa") exists in the graph. KGEs have been proven effective in a variety of web-related downstream tasks, including, for instance, predicting relationship(s) among entities. However, the problem of anticipating the performance of a given KGE in a certain downstream task and locally to a specific individual triple, has not been tackled so far.

In this paper, we fill this gap with *ReliK*, a **Reli**ability measure for **K**GEs. *ReliK* relies solely on KGE embedding scores, is task- and KGE-agnostic, and requires no further KGE training. As such, it is particularly appealing for semantic web applications which call for testing multiple KGE methods on various parts of the KG and on each individual downstream task. Through extensive experiments, we attest that *ReliK* correlates well with both common downstream tasks, such as tail/relation prediction and triple classification, as well as advanced downstream tasks, such as rule mining and question answering, while preserving locality.

## ACM Reference Format:
Anonymous Author(s). 2024. *ReliK*: A Reliability Measure for Knowledge Graph Embeddings. In *Proceedings of (TheWebConf 24)*. ACM, New York, NY, USA, 10 pages. https://doi.org/XXXXXXX.XXXXXXX

## 1 INTRODUCTION

*Knowledge graphs* (KGs) are sets of *facts* (i.e., *triples* such as "da Vinci," "painted," "Mona Lisa") that interconnect *entities* ("da Vinci," "Mona Lisa") via *relationships* ("painted") [18, 41]. Entities and relationships correspond to nodes and (labeled) edges of the KG, respectively (Figure 2). *Knowledge graph embeddings* (KGEs) [39] are popular techniques to generate a vector representation for entities and relationships of a KG. A KGE is computed by optimizing a *scoring function* that provides an *embedding score* as an indication of whether a triple actually exists in the KG. KGEs have been extensively used as a crucial building block of state-of-the-art methods for a variety of *downstream tasks* commonly carried out on the Web, such as knowledge completion [40], whereby a classifier is trained on the embeddings to predict the existence of a triple; or head/tail prediction [22], which aims to predict entities of a triple, as well as more advanced ones, including rule mining [43], query answering [42], and entity alignment [4, 19, 45, 46].

**Motivation.** So far, the choice of an appropriate KGE method has depended on the downstream task, the characteristics of the input KG, and the computational resources. The existence of many

different scoring functions, including linear embeddings [7], bilinear [43], based on complex numbers [33], or projections [9] further complicates this choice. Alas, the literature lacks a unified measure to quantify how *reliable* the performance of a KGE method can be for a certain task *beforehand*, without performing such a potentially slow task. Furthermore, KGE performance on a specific downstream task is typically assessed in a *global* way, that is, in terms of how accurate a KGE method is for that task on *the entire KG*. However, the performance of KGEs for several practical applications (e.g., knowledge completion [40]) typically varies across the parts of the KG. This requires carrying out a performance assessment of KGE *locally* to specific parts of the KG, rather than globally.

**Contributions.** We address all the above shortages of the state of the art in KGE and introduce *ReliK* (**Reli**ability for **K**GEs), a simple, yet principled measure that quantifies the *reliability* of how a KGE will perform on a certain downstream task in a specific part of the KG, without executing that task or (re)training that KGE. To the best of our knowledge, no measure like *ReliK* exists in the literature. *ReliK* relies exclusively on embedding scores as a black box, particularly on the ranking determined by those scores (rather than the scores themselves). Specifically, it is based on the relative ranking of existing KG triples with respect to non-existing ones, in the target part of the KG. As such, *ReliK* is agnostic to both (1) the peculiarities of a specific KGE and (2) the KG at hand, and (3) it needs no KGE retraining. Also, (4) *ReliK* is task-agnostic: in fact, its design principles are so general that it is inherently well-suited for a variety of downstream tasks (see Section 3 for more details, and Section 4 for experimental evidence). Finally, (5) *ReliK* exhibits the locality property, as its computation and semantics can be tailored to a specific part of the KG. All in all, therefore, our *ReliK* measure is fully compliant with all the requirements discussed above. Note that *ReliK* can be used also to evaluate the utility of a KGE for a downstream task, even when (for privacy or other reasons) we only have access to the embedding and not to the original KG.

*ReliK* is simple, intuitive, and easy-to-implement. Despite that, its exact computation requires processing all the possible combinations of entities and relationships, for every single fact of interest. Thus, computing *ReliK* exactly on large KGs and/or large target subgraphs may be computationally too heavy. This is a major technical challenge, which we address by devising approximations to *ReliK*. Our approximations are shown to be theoretically solid (Section 3.2) and perform well empirically (Section 4.1).

**Advanced downstream tasks.** Apart from experimenting with *ReliK* in basic downstream tasks, such as entity/relation prediction or triple prediction, we also showcase *ReliK* on two advanced downstream tasks, to fully demonstrate its general applicability. The first is *query answering*, which finds answers to complex logical queries over KGs. The second, *rule mining*, deduces logic rules, with the purpose of cleaning the KG from spurious facts or expanding the information therein. Rule mining approaches rely on a confidence statistical measure that depends on the quality of the data itself. By

---
*TheWebConf 24, May 13–17, 2024, Singapore*
2024. ACM ISBN 978-1-4503-XXXX-X/18/06...$15.00
https://doi.org/XXXXXXX.XXXXXXX

computing the confidence on a ground truth, we show that *ReliK* identifies more trustworthy rules.

**Relevance.** *ReliK* is particularly amenable to semantic web applications, for instance by providing a local means to study the semantics associated with a specific's entity embedding [27] or by offering an efficient tool for knowledge completion [44].

**Summary and outline.** To summarize, our contributions are:

- We fill an important gap of the state of the art in KGE (Section 2) by tackling for the first time the problem of assessing the reliability of KGEs (Section 3).
- We devise *ReliK*, the first reliability measure for KGEs, which possesses important characteristics of generality, simplicity, and soundness (Section 3.1).
- We devise efficient, yet theoretically solid approximation techniques for estimating *ReliK* (Section 3.2).
- We perform extensive experiments to show that *ReliK* correlates with several common downstream tasks, it complies well with the locality property, and its approximate computation is efficient and effective (Section 4).
- We additionally showcase *ReliK* in two advanced downstream tasks, question answering and rule mining (Section 4.3).

## 2 PRELIMINARIES

A *knowledge graph* (KG) $\mathcal{K} : \langle \mathcal{E}, \mathcal{R}, \mathcal{F} \rangle$ is a triple consisting of a set $\mathcal{E}$ of $n$ entities, a set $\mathcal{R}$ of relationships, and a set $\mathcal{F} \subset \mathcal{E} \times \mathcal{R} \times \mathcal{E}$ of $m$ facts. A *fact* is a *triple* $x_{hrt} = (h, r, t)$[1], where $h \in \mathcal{E}$ is the *head*, $t \in \mathcal{E}$ is the *tail*, and $r \in \mathcal{R}$ is the *relationship*. For instance, entities "Leonardo da Vinci" and "Mona Lisa," and relationship "painted" form the triple ("Leonardo da Vinci," "painted," "Mona Lisa"). The set $\mathcal{F}$ of facts form an edge-labeled graph whose nodes and labeled edges correspond to entities and relationships, respectively. We say a triple $x_{hrt}$ is "positive" if it actually exists in the KG (i.e., $x_{hrt} \in \mathcal{F}$), "negative" otherwise (i.e., $x_{hrt} \notin \mathcal{F}$). KGs are also known as knowledge bases [13], information graphs [23], or heterogeneous information networks [31].

**Knowledge graph embedding.** A *KG embedding* (KGE) [2, 22, 39] is a representation of entities and relationships in a $d$-dimensional ($d \ll |\mathcal{E}|$) space, typically, the real $\mathbb{R}^d$ space or the complex $\mathbb{C}^d$ space. For instance, TransE [7] represents a triple $x_{hrt}$ as entity vectors $\mathbf{e}_h, \mathbf{e}_t \in \mathbb{R}^d$ and relation vector $\mathbf{e}_r \in \mathbb{R}^d$, and DistMult [43] represents the relationship as a matrix $\mathbf{W}_r \in \mathbb{R}^{d \times d}$. Although KGEs can differ (significantly) from one another in their definition, a common key aspect of all KGEs is that they are typically defined based on a so-called *embedding scoring function* or simply *embedding score*. This is a function $s : \mathcal{E} \times \mathcal{R} \times \mathcal{E} \to \mathbb{R}$, which quantifies how likely a triple $x_{hrt} \in \mathcal{E} \times \mathcal{R} \times \mathcal{E}$ exists in $\mathcal{K}$ based on the embeddings of its head ($h$), relationship ($r$), and tail ($t$). Specifically, the higher $s(x_{hrt})$, the more likely the existence of $x_{hrt}$. For instance, TransE's embedding score $s(x_{hrt}) = -\|\mathbf{e}_h + \mathbf{e}_r - \mathbf{e}_t\|$ represents the ($\ell_1$ or $\ell_2$) distance between the "translation" from $h$'s embedding to $t$'s embedding through $r$'s embedding [7].

KGEs are typically learned through a training process that optimizes (e.g., via gradient descent) a loss function defined based on the embedding score. This training process can be computationally

---

[1] We use "fact" and "triple" interchangeably throughout the paper.

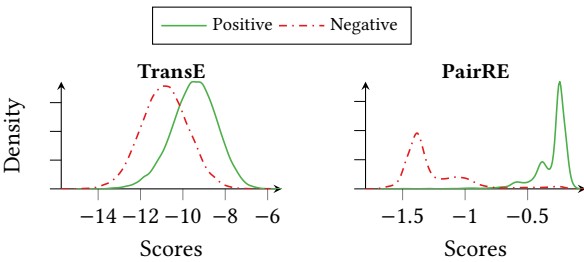

**Figure 1: Distribution of the embedding scores for positive (i.e., existing) and negative (i.e., non-existing) triples on CodexSmall dataset (cf. Section 4), with TransE [7] and PairRE [9] KGE methods. Although scores and distributions are different, positive and negative triples are well separated.**

expensive, especially if it has to be repeated for multiple KGEs. KGEs learned this way are shown to be effective for a number of downstream tasks [22], such as predicting the existence of a triple, but do not offer any prior indication on their performance [20]. Moreover, existing benchmarks [2] show global performance on the entire graph rather than *local* on subgraphs. To this end, in this work, we provide an answer to the following key question:

> MAIN QUESTION. *Is there a measure that provides a prior indication of the performance of a KGE on a specific subgraph?*

## 3 KGE RELIABILITY

A good measure of performance of a KGE should support a number of tasks, from node classification, to link prediction, as well as being unprejudiced towards the data and the KGE model itself. In other words, we would like a measure of *reliability* that properly assesses how the embedding of a triple would perform on certain tasks and data, without knowing them in advance. More specifically, the main desiderata of a proper KGE reliability measure are as follows.

**(R1) Embedding-agnostic.** It should be independent of the specific KGE method. This is to have a measure fully general.

**(R2) Learning-free.** It should require no further KGE training. This is primarily motivated by efficiency, but also for other reasons, such as privacy or unavailability of the data used for KGE training.

**(R3) Task-agnostic.** It should be independent of the specific downstream task. In other words, it should be able to properly anticipate the performance of a KGE in general, for *any* downstream task. Again, like **(R1)**, this is required for the generality of the measure.

**(R4) Locality.** It should be a good predictor of KGE performance *locally* to a given triple, that is, in a close surrounding neighborhood of that triple. This is important, as a KGE model may be more or less effective based on the different parts of the KG it is applied to. Thus, assessing how KGEs perform in different parts of the KG would allow for better use and combine them in downstream tasks.

### 3.1 The Proposed *ReliK* Measure

**Design principles.** Defining a reliability measure that complies with the aforementioned requirements is an arduous streak. First, the various KGE methods consider different objectives. Second, downstream tasks often combine embeddings in different ways. For instance, even though head or tail predictions predict a single

vector, triple classification combines head, tail, and relationship vectors. Third, the embedding scores are in general incomparable across the KGEs.

To fulfil **(R1)** and **(R2)**, the KGE reliability measure should not engage with the internals of the computation of KGEs. Thus, we need to treat the embeddings as vectors and the embedding score as a black-box function that provides only an indication of the actual existence of a triple. Though the absolute embedding scores are incomparable to one another, we observe that the distribution of positive and negative triples is significantly different (Figure 1). Specifically, the *relative ranking* of a positive triple is higher than that of a negative. This leads to the following main observation.

OBSERVATION 1. *A KGE reliability measure that uses the position of a triple relative to other triples via a ranking defined based on the embedding score fulfills **(R1)** and **(R2)**.*

Furthermore, comparing a triple to all other (positive or negative) triples might be ineffective. For instance, if we assume that our measure of reliability is solely based on the separation between positive and negative triples, we will conclude from Figure 1 that PairRE [9] performs well for all the tasks, which is not the case. This is because the absolute score does not provide an indication of performance. We thus advocate that a local approach that considers triples relative to a neighborhood is more appropriate, and propose a measure that fulfils **(R4)**. The soundness of **(R4)** is better attested in our experiments in Section 4.

Finally, to meet **(R3)**, the KGE reliability measure should not exploit any peculiarity of a downstream task in its definition. Indeed, this is accomplished by our measure, as we show next.

**Definition.** For a triple $x_{hrt} = (h, r, t)$ we compute the neighbor set $\mathcal{N}^-(h)$ of all possible negative triples, i.e., triples with head $h$ that do not exist in $\mathcal{K}$. Similarly, we compute $\mathcal{N}^-(t)$ for tail $t$. We define the *head-rank h* of a triple $x_{hrt}$ as the position of the triple in the rank obtained using score $s$ for a specific KGE relative to all the negative triples having head $h$.

$$rank_H(x_{hrt}) = |\{x \in \mathcal{N}^-(h) : s(x) > s(x_{hrt})\}| + 1$$

The *tail-rank $rank_T(x_{hrt})$* for tail $t$ is defined similarly.

Our reliability measure, *ReliK*, for a triple $x_{hrt}$ is ultimately defined as the average of the reciprocal of the head- and tail-rank

$$ReliK(x_{hrt}) = \frac{1}{2}\left(\frac{1}{rank_H(x_{hrt})} + \frac{1}{rank_T(x_{hrt})}\right) \quad (1)$$

*ReliK* can easily be extended from single triples to subgraphs by computing the average reliability among the facts in the subgraph. Specifically, we define the *ReliK* score of a set $S \subseteq \mathcal{F}$ of triples as

$$ReliK(S) = \frac{1}{|S|}\sum_{x_{hrt} \in S} ReliK(x_{hrt}). \quad (2)$$

**Rationale.** *ReliK* ranges from (0, 1], with higher values corresponding to better reliability. In fact, the lower the head-rank $rank_H(x_{hrt})$ and/or tail-rank $rank_T(x_{hrt})$, the better the ranking of $x_{hrt}$ induced by the underlying embedding scores, relatively to the non-existing triples in $x_{hrt}$'s neighborhood, complies with the actual existence of $x_{hrt}$ in the KG.

It is easy to see that *ReliK* achieves **(R1)** and **(R2)** by relying on the relative ranking rather than the absolute scores. It also fulfils

**(R3)** as it involves no downstream tasks at all, and **(R4)** as it is based on the local (i.e., 1-hop) neighborhood of a target triple.

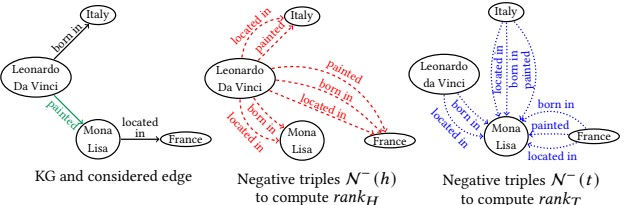

**Figure 2: Constituents of *ReliK* on an example KG.**

Figure 2 provides an example of the computation of *ReliK* for the triple $x_{hrt}$ = ("Leonardo da Vinci", "painted", 'Mona Lisa"). The $\mathcal{N}^-(h)$ is depicted as the red (dashed) edges and $\mathcal{N}^-(t)$ in blue (dotted). To compute *ReliK* on an embedding, we compute the embedding score $s$ of ("Leonardo da Vinci", "painted", "Mona Lisa") and rank it according to the triples in $\mathcal{N}^-(h)$ and $\mathcal{N}^-(t)$.

## 3.2 Efficiently computing *ReliK*

Computing *ReliK* (Eq. (1)) takes $O(|\mathcal{E}| \cdot |\mathcal{R}|)$ time, as it needs to scan the whole negative neighborhood of the target triple. For large KGs, repeating this for a (relatively) high number of triples may be computationally too heavy. For this purpose, here we focus on approximate versions of *ReliK*, which properly trade off between accuracy and efficiency.

The main intuition behind the *ReliK* approximation is that the precise ranking of the various potential triples is not actually needed. Rather, what it matters is *just the number* of those triples that exhibit a higher embedding score than the target triple. This observation leads to two approaches. In both of them, we sample a random subset of negative triples. In the first approach, we compute $ReliK_{LB}$, a lower bound to *ReliK*, by counting the negative triples in the sample that have a lower embedding score than the target triple and pessimistically assuming that all the other triples not in the sample have higher scores. In the second approach, we estimate $ReliK_{Apx}$ by evaluating the fraction of triples in the sample that have a higher score than the triple under consideration and then scaling this fraction to the total number of negative triples. Next, we provide the details of these two approaches.

Let $S_H$ be a random subset of $k$ elements selected without replacement independently and uniformly at random from the negative neighborhood $\mathcal{N}^-(h)$ of the head $h$ of a triple $x_{hrt}$. The size $|S_H|$ trades off between efficiency and accuracy of the estimator, and it may be defined based on the size of $\mathcal{N}^-(h)$. Define also

$$rank_H^S(x_{hrt}) = |\{x \in S_H : s(x) > s(x_{hrt})\}| + 1,$$

to be the rank of the score $s(x_{hrt})$ that the KGE assigns to $x_{hrt}$, among all the triples *in the sample*. We similarly compute $S_T$ and RANK$_T^S$ for tail's neighborhood $\mathcal{N}^-(t)$.

**$ReliK_{LB}$ estimator.** The sampled triples with lower score than $s(x_{hrt})$ are fewer than all such negative triples, that is,

$$|S_H| - rank_H^S(x_{hrt}) \leq |\mathcal{N}^-(h)| - rank_H(x_{hrt}),$$

or, equivalently,

$$rank_H(x_{hrt}) \leq rank_H^S(x_{hrt}) + |\mathcal{N}^-(h)| - |S_H|. \quad (3)$$

Analogously, the observation holds for $S_T$

$$rank_T(x_{hrt}) \leq rank_T^S(x_{hrt}) + |\mathcal{N}^-(t)| - |S_T| . \quad (4)$$

We therefore define our $ReliK_{LB}$ estimator as

$$ReliK_{LB}(x_{hrt}) = \frac{1}{2}\left( \frac{1}{rank_H^S(x_{hrt}) + |\mathcal{N}^-(h)| - |S_H|} \right.$$
$$\left. + \frac{1}{rank_T^S(x_{hrt}) + |\mathcal{N}^-(t)| - |S_T|} \right), \quad (5)$$

From Eqs. (3) and (4), it holds that

$$ReliK_{LB}(x_{hrt}) \leq ReliK(x_{hrt}).$$

**$ReliK_{Apx}$ estimator.** As for our second estimator, we define it as

$$ReliK_{Apx} = \frac{1}{2}\left( \frac{1}{rank_H^S(x_{hrt}) \frac{|\mathcal{N}^-(h)|}{|S_H|}} + \frac{1}{rank_T^S(x_{hrt}) \frac{|\mathcal{N}^-(t)|}{|S_T|}} \right). \quad (6)$$

In words, we simply scale up the rank induced by the sample to the entire set of negative triples.

**Theoretical characterization of $ReliK_{Apx}$.** Note that by Jensen's inequality [21], we have that

$$\mathbb{E}\left[ \frac{1}{rank_H^S(x_{hrt}) \frac{|\mathcal{N}^-(h)|}{|S_H|}} \right] \geq \frac{1}{\mathbb{E}\left[ rank_H^S(x_{hrt}) \frac{|\mathcal{N}^-(h)|}{|S_H|} \right]}$$
$$= \frac{1}{\mathbb{E}[rank_H^S(x_{hrt})] \frac{|\mathcal{N}^-(h)|}{|S_H|}} = \frac{1}{rank_H(x_{hrt})},$$

where $\mathbb{E}[\cdot]$ denotes mathematical expectation. This holds since

$$\mathbb{E}[rank_H^S(x_{hrt})] = |S_H| \cdot \frac{rank_H(x_{hrt})}{|\mathcal{N}^-(h)|},$$

given that for each element $x \in S_H$, the probability to have a score $s(x) > s(x_{hrt})$ is

$$\frac{rank_H(x_{hrt})}{|\mathcal{N}^-(h)|}.$$

We argue similarly for the tail and, therefore, we finally obtain

$$\mathbb{E}[ReliK_{Apx}(x_{hrt})] \geq ReliK(x_{hrt}).$$

In other words, $ReliK_{Apx}$ is, in expectation, an *upper bound* of $ReliK$.

**Quality of $ReliK_{Apx}$ approximation.** $ReliK_{Apx}$ is a randomized approximate based on Bernoulli trials. To see that, let us consider each negative triple in $\mathcal{N}^-(h)$ a sample from an i.i.d. Bernoulli variable with probability $p = (rank_H(x_{hrt}) - 1)/|\mathcal{N}^-(h)|$. In other words, we assume that each negative triple $x$ with score $s(x) > s(x_{hrt})$ is assigned a binary random variable $B_x = 1$ and the rest 0. As such, $rank_H(x_{hrt}) - 1 = \sum_{x \in \mathcal{N}^-(h)} B_x$ is a sum of Bernoulli variables which are distributed as a binomial random variable $f(k; |\mathcal{N}^-(h)|, p)$. Thus, our $ReliK_{Apx}$ for a sample of size $k$ bounds the errors within

$$\left| \frac{k}{|\mathcal{N}^-(h)|} - p \right|$$

The same reasoning and bound hold for the tail $t$.

**Algorithms.** Algorithm 1 shows the steps to compute $ReliK_{LB}$ and $ReliK_{Apx}$. Initially, in Line 1, we sample, uniformly at random, $k$ negative triples from the head neighborhood and the tail neighborhood. Note that we can save computation time by first filtering the

---

**Algorithm 1** compute $ReliK_{LB}$ or $ReliK_{Apx}$

**Input:** KG $\mathcal{K} : \langle \mathcal{E}, \mathcal{R}, \mathcal{F} \rangle$, triple $x_{hrt} = (h, r, t) \in \mathcal{F}$, embedding score function $s : \mathcal{E} \times \mathcal{R} \times \mathcal{E} \to \mathbb{R}$, sample size $k \in \mathbb{N}$
**Output:** $ReliK_{LB}(x_{hrt})$ (Eq. (5)) or $ReliK_{Apx}(x_{hrt})$ (Eq. (6))
1: $S_H \leftarrow$ sample $k$ triples from $\mathcal{N}^-(h)$; $S_T \leftarrow$ sample $k$ triples from $\mathcal{N}^-(t)$
2: $rank_H \leftarrow 1$; $rank_T \leftarrow 1$
3: **for** $x_{h'r't'} \in S_H \cup S_T$ **do**
4:   **if** $s(x_{hrt}) < s(x_{h'r't'})$ **then**
5:     **if** $h' = h$ **then**
6:       $rank_H \leftarrow rank_H + 1$
7:     **if** $t' = t$ **then**
8:       $rank_T \leftarrow rank_T + 1$
9: **return** $\frac{1}{2}\left( \frac{1}{rank_H + |\mathcal{N}^-(h)| - |S_H|} + \frac{1}{rank_T + |\mathcal{N}^-(t)| - |S_T|} \right)$ for $ReliK_{LB}$

  or $\frac{1}{2}\left( \frac{1}{rank_H \frac{|\mathcal{N}^-(h)|}{|S_H|}} + \frac{1}{rank_T \frac{|\mathcal{N}^-(t)|}{|S_T|}} \right)$ for $ReliK_{Apx}$

---

triples in $S_H \cup S_T$ by score (Line 4), i.e., considering only those with score higher than the input triple $x_{hrt}$, and then check whether a triple in $S_H \cup S_T$ has either the head (Line 5) or the tail (Line 7) in common with $x_{hrt}$ to update the corresponding rank.

**Time complexity.** Algorithm 1 runs in $O(k)$ time. This corresponds to the time needed for the sampling step in Line 5, which can easily be accomplished linearly in the number of samples, without materializing the negative neighborhoods. The sample size $k$ trades off between accuracy and efficiency of the estimation. Section 4.1 shows that $ReliK_{Apx}$ approximation with 20% sample size is $2.5\times$ faster than $ReliK$ with only 0.002 Mean Squared Error (MSE). As such, $ReliK_{Apx}$ is our method of reference in the experiments.

## 4 EXPERIMENTAL EVALUATION

We evaluate $ReliK$ on four downstream tasks, six embeddings, and six datasets. We report the correlation with $ReliK$ and the performance of ranking tasks (Section 4.2) and show that $ReliK$ can identify correct query answers as well as mine rules with higher confidence than existing methods (Section 4.3).

| method | space | | | score |
|---|---|---|---|---|
| | set | entity | relation | |
| TransE [7] | $\mathbb{R}$ | $O(n)$ | $O(n)$ | $-\|\mathbf{e}_h + \mathbf{e}_r - \mathbf{e}_t\|_p$ |
| DistMult [43] | $\mathbb{R}$ | $O(n)$ | $O(n)$ | $\mathbf{e}_h^\top \text{diag}(\mathbf{W}_r) \mathbf{e}_t$ |
| RotatE [33] | $\mathbb{C}$ | $O(n)$ | $O(n)$ | $-\|\mathbf{e}_h \circ \mathbf{e}_r - \mathbf{e}_t\|$ |
| PairRE [9] | $\mathbb{R}$ | $O(n)$ | $O(n)$ | $-\|\mathbf{e}_h \circ \mathbf{e}_{rh} - \mathbf{e}_t \circ \mathbf{e}_{rt}\|$ |
| ComplEx [38] | $\mathbb{C}$ | $O(n)$ | $O(n)$ | $Re(\langle \mathbf{e}_r, \mathbf{e}_h, \overline{\mathbf{e}_t} \rangle)$ |
| ConvE [14] | $\mathbb{R}$ | $O(n)$ | $O(n)$ | $f(vec(f([\overline{\mathbf{e}_h}; \overline{\mathbf{e}_r}] * \omega))\mathbf{W})\mathbf{e}_t$ |

**Table 1: Characteristics of the considered embeddings.**

**Embeddings.** We include six established KGE methods, representative of the four major embedding families (see Section 5). Table 1 shows the embeddings in our evaluation, the embedding space, and the embedding score function. A detailed description of the embeddings is in Section A.1 in the appendix.

**Datasets.** We perform experiments on six KGs with different characteristics, shown in Table 2.

- **Countries** [8] is a small KG created from geographical locations, where entities are continents, subcontinents and countries, and edges containment or geographical neighborhood.
- **FB15k237** [37] is a sample of Freebase KG [6] covering encyclopedic knowledge consisting of 237 relations, 15$k$ entities and 310$k$

| dataset | $|\mathcal{E}|$ | $|\mathcal{R}|$ | $|\mathcal{F}|$ | Task |
|---|---|---|---|---|
| Countries | 271 | 2 | 1 158 | Approximation |
| FB15k237 | 14 505 | 237 | 310 079 | Ranking / Classification / Querying |
| Codex-S | 2 034 | 42 | 36 543 | Ranking / Classification |
| Codex-M | 17 050 | 51 | 206 205 | Ranking / Classification |
| Codex-L | 77 951 | 69 | 612 437 | Ranking / Classification |
| YAGO2 | 834 750 | 36 | 948 358 | Rule Mining |

**Table 2: Characteristics of the KGs; number of entities $|\mathcal{E}|$; number of relationships $|\mathcal{R}|$; number of facts $|\mathcal{F}|$; task.**

facts. FB15k237 is a polished and corrected version of FB15k [7] constructed to circumvent data leakage. The dataset contains Freebase entities with more than 100 mentions and with reference in Wikilinks database.

- **Codex** [29] is a collection of three datasets of incremental size, Codex-S ($2k$ entities, $36k$ triples), Codex-M ($17k$ entities, $200k$ facts), and Codex-L ($78k$ entities, $610k$ facts) extracted from Wikidata and Wikipedia. Codex collection explicitly encourages entity and content diversity to overcome the limitations of FB15k.
- **YAGO** [32] is an open-source KG automatically extracted from Wikidata with an additional ontology from schema.org. We use YAGO2 [17], which comprises 834k entities and 948k facts.

**Experimental setup.** We implement our approximate and exact *ReliK* in Python v3.9.13.[2] We train the embedding using the Pykeen library v1.10.1,[3] with default parameters besides the embedding dimension $dim = 50$ and training loop sLCWA. We run our experiments on a Linux Ubuntu 4.15.0-202 machine with 48 cores Intel(R) Xeon(R) Silver 4214 @ 2.20GHz, 128GB RAM and an NVIDIA GeForce RTX 2080 Ti GPU. We report an average of 5 experiments using 5-fold cross validation with 80-10-10 split.

**Summary of experiments.** We evaluate *ReliK* on several downstream tasks and setups. We first show in Section 4.1 that our approximate *ReliK*$_{Apx}$ outperforms the simpler *ReliK*$_{LB}$ lower-bound approximation and achieves a good tradeoff between quality and speed. We then show in Section 4.2 that *ReliK* correlates with common ranking tasks, such as tail and relation prediction, as well as classification tasks and validate the claim that *ReliK* is a local measure. In Section 4.3 we present the more complex tasks of query answering and mining logic rules on KGs. To summarize, we evaluate *ReliK* on the following downstream tasks:

- **(T1)** Ranking tasks, tail and relation prediction
- **(T2)** Classification task, triple classification
- **(T3)** Query answering task
- **(T4)** Rule mining application

## 4.1 Approximation Quality

We start by showing that *ReliK*$_{Apx}$ runs as fast as *ReliK*$_{LB}$ while being more accurate. We report time and mean squared error (MSE) with respect to the exact *ReliK* measure for *ReliK*$_{Apx}$ and *ReliK*$_{LB}$. Computing *ReliK* is infeasible in datasets with more than a few hundred entities. Hence, we limit our analysis to the entire Countries dataset for which we can compute *ReliK* exactly.

Figure 3 reports the results in terms of seconds and MSE at increasing sample size $k = |S|$. Both *ReliK*$_{LB}$ and *ReliK*$_{Apx}$ incur the

[2]Code available at: https://anonymous.4open.science/r/Anon-6405
[3]https://pykeen.readthedocs.io/en/stable/

same time, because of the fact that both require to sample $k$ negative triples and compute the score on the sample. On the other hand, when the sample size is more than 80% of all the negative triples, as the sampling time dominates the computation of *ReliK*$_{LB}$ and *ReliK*$_{Apx}$, *ReliK* becomes faster. *ReliK*$_{Apx}$ rapidly reduces the error and stabilizes at around 40% of the sample size, whereas *ReliK*$_{LB}$ exhibits a steadily larger error than *ReliK*$_{Apx}$. The current results show the effectiveness of the results in an unparallelized setting; yet, we note that the sampling process can be easily parallelized by assigning each sample to a separate thread.

In terms of quality, *ReliK*$_{Apx}$ exhibits minimal MSE (<0.005) with as little as 10% of the sample size, being 3 times faster than *ReliK*. Thus, even though the exact *ReliK* is feasible for small datasets or subgraphs, *ReliK*$_{Apx}$ offers a good approximation with significant speedup. On the next experiments, we set $k$ to 10% of all the negative triples and report results for *ReliK*$_{Apx}$.

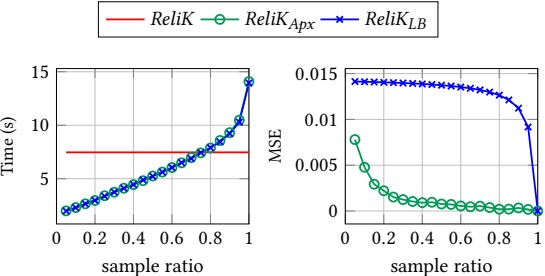

**Figure 3: Comparing *ReliK*$_{Apx}$ and *ReliK*$_{LB}$ with exact *ReliK* in time (left) and Mean Squared Error (right) vs sample to data size ratio on Countries dataset and TransE embeddings.**

## 4.2 Common Downstream Tasks

We test *ReliK* on the ability to anticipate the results of common tasks for KGEs [22, 39]. We measure the statistical significance of Pearson correlation among two ranking tasks, tail and relation prediction, and the triple classification task. To evaluate *ReliK* on different areas of the graph and different graph topologies, we sample random subgraphs of Codex-S with 60 nodes by initially selecting a starting node uniformly at random and then including nodes and edges by random walk with restart [36] with restart probability $1 - \alpha = 0.2$, until the subgraph comprises 60 nodes. For Codex-M and Codex-L we use size 100 and for FB15k237 we use 200 nodes. We report the average *ReliK* on 100 random subgraphs on the Codex-S, Codex-M, Codex-L, and FB15k237 datasets.

**Ranking tasks (T1).** In the first experiments, we measure the Pearson correlation between *ReliK* and the performance on ranking tasks with mean reciprocal rank (MRR) [11]. The first task, *tail prediction* [7, 9, 33], assesses the ability of the embedding to predict the tail given the head and the relation, thus answering the query $(h, r, ?)$ where the tail is unknown. The second task, *relation prediction*, assesses the ability of the embedding to predict the undisclosed relation of a triple $(h, ?, t)$. The common measure used for tail and relation prediction is MRR, which provides an indication of how close to the top the score ranks the correct tail (or relation). Consistently with previous approaches [7, 9, 33], we employ the filtered approach in which we consider for evaluation only negative triples that do not appear in either the train, test, or validation set. Table 3 reports the correlations alongside the statistical significance

in terms of the p-value. We marked in red, high p-values ($> 0.05$) that suggest no correlation and person score values that are for inverse correlation. Generally, *ReliK* exhibits significant correlation across embeddings and tasks. Noteworthy, even though *ReliK* (see Eq. (1)) does not explicitly target tail or head rankings by including both, we observe significant correlation on tail prediction in most embeddings and datasets. Because of the considerable training time, we only report results for RotatE on Codex-S. We complement our analysis with correlation plots in Figure 4 for Codex-S; in most cases, we observe a clear correlation. Comparing the actual results of the various tasks, it is also clear in most cases in which we do not have correlation, that the results are too close to distinguish; for example, ComplEx having only result close to 0. In such cases, *ReliK* indicates that the embedding needs further tuning.

| KGE | Tail (MRR) | | Relation (MRR) | | Classific. (Acc.) | |
|---|---|---|---|---|---|---|
| | Pearson | p-value | Pearson | p-value | Pearson | p-value |
| **Codex-S** | | | | | | |
| TransE | 0.23 | 0.02 | 0.93 | $2.17e^{-44}$ | 0.37 | $1.42e^{-4}$ |
| DistMult | 0.16 | 0.12 | 0.85 | $2.03e^{-29}$ | 0.69 | $2.21e^{-15}$ |
| RotatE | 0.35 | 0.0003 | 0.89 | $7.92e^{-37}$ | −0.24 | 0.02 |
| PairRE | 0.86 | $7.29e^{-31}$ | 0.91 | $2.36e^{-39}$ | 0.09 | 0.37 |
| ComplEx | 0.14 | 0.17 | 0.63 | $2.22e^{-12}$ | −0.06 | 0.57 |
| ConvE | −0.396 | $6.61e^{-5}$ | 0.89 | $4.92e^{-37}$ | 0.10 | 0.30 |
| **Codex-M** | | | | | | |
| TransE | 0.90 | $2.70e^{-37}$ | 0.97 | $9.07e^{-63}$ | 0.53 | $1.93e^{-08}$ |
| DistMult | 0.22 | 0.04 | 0.89 | $8.37e^{-32}$ | 0.60 | $5.12e^{-10}$ |
| RotatE | – | – | – | – | – | – |
| PairRE | 0.06 | 0.58 | 0.98 | $1.05e^{-74}$ | −0.12 | 0.23 |
| ComplEx | −0.33 | $8.92e^{-4}$ | 0.36 | $2.01e^{-4}$ | 0.15 | 0.13 |
| ConvE | −0.22 | 0.03 | 0.99 | $3.86e^{-96}$ | −0.02 | 0.84 |
| **Codex-L** | | | | | | |
| TransE | 0.83 | $1.13e^{-26}$ | 0.97 | $3.812e^{-64}$ | 0.63 | $2.54e^{-12}$ |
| DistMult | 0.49 | $2.10e^{-07}$ | 0.78 | $4.68e^{-22}$ | 0.60 | $3.74e^{-11}$ |
| RotatE | – | – | – | – | – | – |
| PairRE | −0.04 | 0.68 | 0.95 | $3.33e^{-52}$ | $-4.47e^{-4}$ | 0.99 |
| ComplEx | 0.82 | $1.03e^{-25}$ | 0.91 | $3.96e^{-39}$ | 0.06 | 0.57 |
| ConvE | 0.59 | $4.26e^{-11}$ | −0.07 | 0.48 | 0.31 | $1.57e^{-3}$ |
| **FB15k237** | | | | | | |
| TransE | 0.24 | 0.02 | 0.86 | $2.83e^{-30}$ | 0.34 | $5.79e^{-4}$ |
| DistMult | −0.05 | 0.65 | 0.64 | $5.57e^{-13}$ | 0.39 | $5.58e^{-05}$ |
| RotatE | – | – | – | – | – | – |
| PairRE | 0.80 | $1.51e^{-23}$ | 0.65 | $1.74e^{-13}$ | 0.08 | 0.44 |
| ComplEx | 0.20 | 0.05 | 0.88 | $3.53e^{-34}$ | 0.14 | 0.18 |
| ConvE | 0.09 | 0.37 | 0.85 | $4.47e^{-30}$ | 0.01 | 0.93 |

**Table 3: Pearson Correlation and statistical significance of *ReliK* for Tail, Relation prediction, and Triple Classification; red indicates cases of less statistically significant correlation, with p-value $> 0.05$, or inverse correlation.**

**Classification task (T2).** In this experiment, we test the correlation between *ReliK* and the accuracy of a threshold-based classifier on the embeddings. The classifier predicts the presence of a triple in the KG if the embedding score is larger than a threshold, a common scenario for link prediction [22]. We tune the threshold on the training set and test it on the test set. Table 3 (right column) reports the correlations and their significance for all datasets and Figure 5 shows the detailed analysis on Codex-S. At close inspection, we

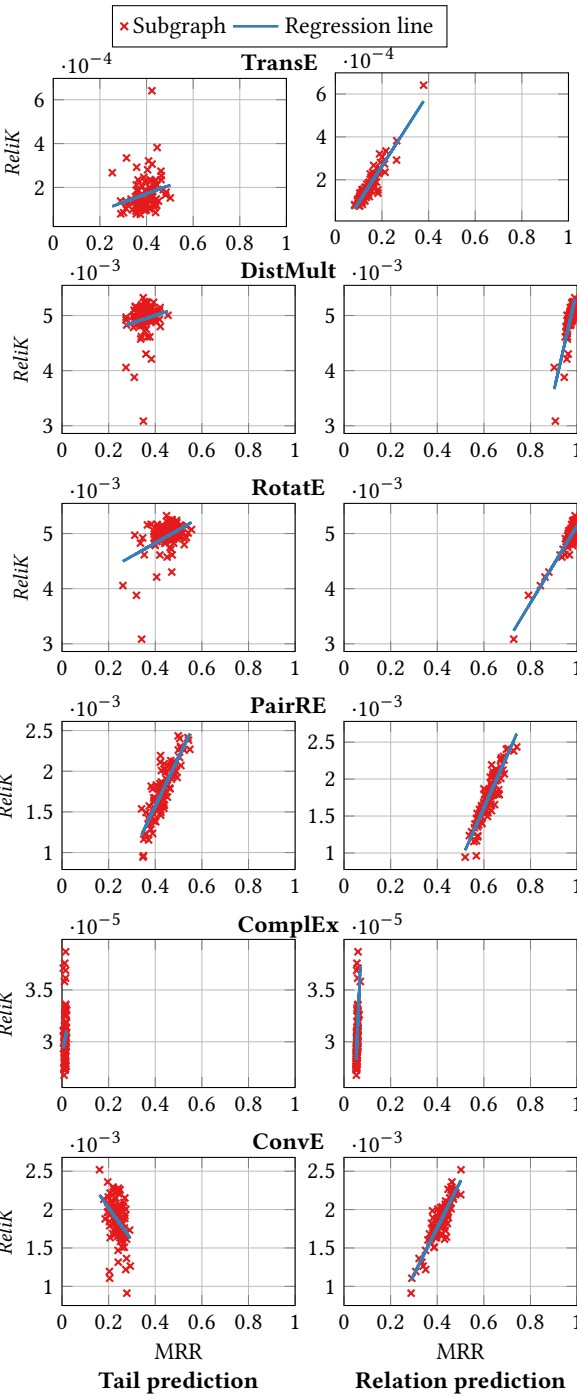

**Figure 4: *ReliK* correlation with MRR on tail prediction (left column) and relation prediction (right column); each point is the *ReliK* score for a subgraph with 60 nodes on Codex-S.**

observe that in cases of unclear correlation, e.g., PairRE, the respective classification results are too close to make out a difference. Those cases notwithstanding, *ReliK* is significantly correlated with accuracy. This result confirms that *ReliK* can serve as a proxy for the quality of complex models trained on embeddings.

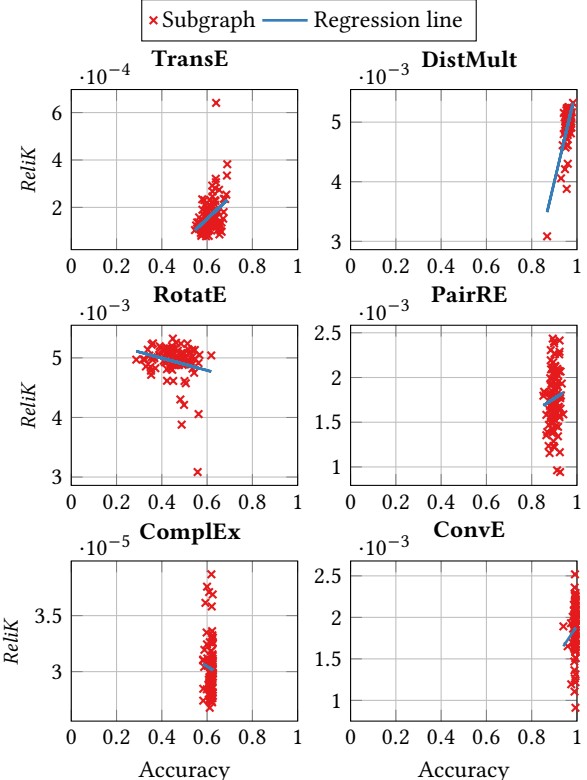

**Figure 5: *ReliK* correlation with Accuracy on triple classification; each point represents the *ReliK* score for a subgraph with 60 nodes on Codex-S.**

**Tuning Subgraph Size.** Next, we analyze how *ReliK* correlates with the tasks presented in Section 4.2 on subgraphs of varying size with the TransE embedding. Figure 6 reports the correlation values for all three tasks, only including those values where the p-value is below 0.05. We observe that *ReliK*'s correlation generally increases with subgraphs of up to 100 nodes on Codex-S. After that point, we note an unstable behavior in all tasks. This is consistent with the assumption that *ReliK* is a measure capturing local reliability. To strike a balance between quality and time we test on subgraphs with 60 nodes for Codex-S in all experiments. Yet, as tasks are of different nature, the subgraph size can be tuned in accordance with the task to provide more accurate results.

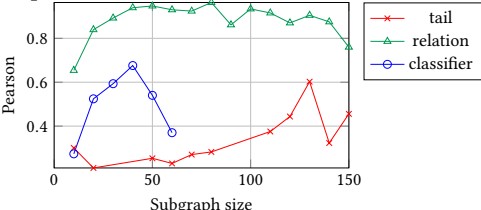

**Figure 6: Pearson correlation on tail and relation prediction and triple classification vs subgraph size on Codex-S.**

### 4.3 Complex Downstream Tasks

We now turn our attention to complex downstream tasks.

**Query answering (T3).** We show how *ReliK* can improve query-answering tasks. Complex logical queries on KGs are working with different query structures. We focus on queries of chaining multiple predictions or having an intersection of predictions, from different query structures that have been described in recent work [3, 28]. We keep the naming convention introduced by Ren and Leskovec [28]. We evaluate a selection of 1000 queries per type ($1p$,$2p$,$3p$,$2i$,$3i$) from their data on the FB15k237 graph.[4] The queries of type $p$ are 1 to 3 hops from a given entity with fixed relation labels that point to a solution, whereas queries of type $i$ are the intersection of 2 or 3 predictions pointing towards the same entity. We evaluate *ReliK* on the ability to detect whether an instance of an answer is true or false. We compute *ReliK* on TransE embeddings trained on the entire FB15k237. Figure 7 shows the average *ReliK* scores for positive and negative answers. *ReliK* clearly discriminates between positive and negative instances, often by a large margin.

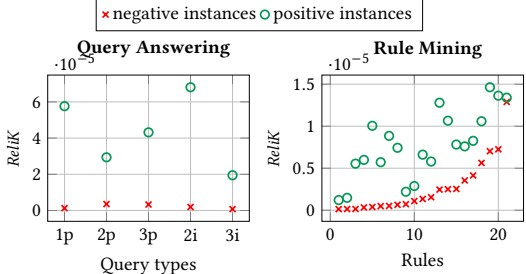

**Figure 7: Comparison between positive instances and negative instances for the query and answer task for FB15k237 (left) and the rule mining task on Yago2 (right).**

**Rule mining (T4).** *ReliK* effectively improves on the rule mining task as well. Rule mining methods [15, 16, 26] automatically retrieve logic rules over KGs having a predefined minimum confidence. A logic rule is an assertion such as $A \implies B$, which states that $B$ follows from $A$. For instance, a rule could imply that all presidents of a country are citizens of the country. An instance of a rule is triples matching $B$, given that $A$ is true. Logic rules are typically harvested with slow exhaustive algorithms similar to the apriori algorithm for association rules [1]. We present two experiments. In the first, we show that *ReliK* can discriminate between true and false instances. In the second, we show that *ReliK* can retrieve all the rules by considering only subgraphs with high *ReliK* score.

**Detecting true instances.** To showcase performance on the downstream task (**T4**), we evaluate *ReliK* on the ability to detect whether an instance of a rule is true or false. This task is particularly important to quantify the real confidence of a rule [24]. To this end, we employ a dataset[5] comprising 23 324 manually annotated instances over 26 rules extracted from YAGO2 using the AMIE [16] and RudiK [26] methods. We compute *ReliK* on TransE embeddings trained on the entire YAGO2. Figure 7 shows the average *ReliK* scores for positive and negative instances. *ReliK* discriminates between positive and negative instances, often by a large margin.

**Rule mining on subgraphs.** In this experiment, we show that *ReliK* identifies the subgraphs with high-confidence rules. To this end, we mine rules with AMIE [15, 16] on Codex-S, and compare with densest subgraphs of increasing size. We construct subgraphs of increasing size by first mining the densest subgraph using Charikar's

---

[4]http://snap.stanford.edu/betae/

[5]https://hpi.de/naumann/projects/repeatability/datasets/colt-dataset.html

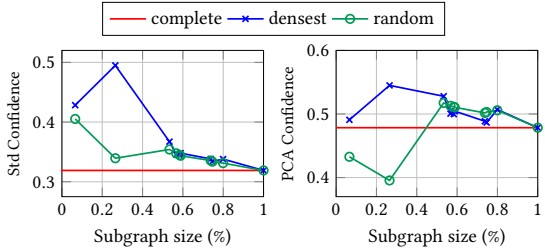

**Figure 8: Std and PCA confidence [16] vs subgraph size for AMIE rules on Codex-S; densest subgraph according to *ReliK*. PCA confidence normalizes the support of a rule *only* by the number of facts which we know to be true *or* consider to be false on a KG *assumed* to be *partially complete* [15, 16].**

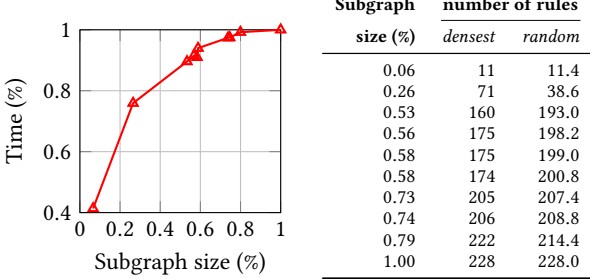

| Subgraph | number of rules | |
|---|---|---|
| size (%) | *densest* | *random* |
| 0.06 | 11 | 11.4 |
| 0.26 | 71 | 38.6 |
| 0.53 | 160 | 193.0 |
| 0.56 | 175 | 198.2 |
| 0.58 | 175 | 199.0 |
| 0.58 | 174 | 200.8 |
| 0.73 | 205 | 207.4 |
| 0.74 | 206 | 208.8 |
| 0.79 | 222 | 214.4 |
| 1.00 | 228 | 228.0 |

**Figure 9: Time to compute AMIE rules vs subgraph size (left) and number of discovered rules (right) on Codex-S.**

greedy algorithm [10] on the weighted graph obtained assigning each edge the *ReliK* score; then, we remove the densest subgraph and repeat the algorithm on the remaining edges, until no edge remains. At each iteration, we mine AMIE rules and compute the standard confidence, as well as confidence by the *partial completeness assumption* (PCA) [15, 16], i.e., the assumption that the database includes either *all* or *none* of the facts about each head entity $h$ by any relationship $r$. In Figure 8 we compare our method with a baseline that extracts random subgraphs of the same size as those computed with our method. The densest subgraph located by *ReliK* finds more rules with higher confidence on as little as 25% of the KG. On the other hand, a random subgraph does not identify any meaningful subgraph. This indicates that *ReliK* is an effective tool for retrieving rules in large graphs. A further analysis in Figure 9 shows that by exploiting *ReliK* we can compute rules 75% of the time. We emphasize though that because rule mining incurs exponential time, the difference between mining rules on the complete graph and on the *ReliK*-subgraph will be more pronounced on graphs larger than Codex-S. As a complement, the table reports the number of rules mined in the entire graph that are discovered by *ReliK* in the subgraph. It is clear, that on 26% of the graph, *ReliK* discovers 1/3 as opposed to only 1/6 discovered by random graphs.

## 5  RELATED WORK

**Knowledge graph embeddings** are commonly used to detect missing triples, correcting errors, or question answering [22, 39]. A number of KGEs appeared in the last few years. The distinctive features among embeddings are the score function and the optimization loss. *Translational embeddings* in the TransE [7] family and the recent PairRE [9] assume that the relationship performs

a translation from the head to the tail. *Semantic embeddings*, such as DistMult [43] or HolE [25] interpret the relationship as a multiplicative operator. *Complex embeddings*, such as RotatE [33] and ComplEx [38] use complex-valued vectors and operations in the complex plane. *Neural-network embeddings*, such as ConvE [14] perform sequences of non-linear operations. While each embedding defines a specific score, *ReliK* is agnostic to the choice of embedding. It is still an open question how well embeddings capture the semantics included in a KG [20]. Our work progresses in that regard by offering a simple local measure to quantify how faithful an embedding represents the information in the data.

**Embedding calibration.** An orthogonal direction to ours is embedding calibration [30, 34]. Calibration methods provide effective ways to improve the existing embeddings on various tasks, by altering the embedding vectors in subspaces with low accuracy [30], by reweighing the output probabilities in the respective tasks [34], or by matrix factorization [12]. On the contrary, *ReliK* does not alter the embeddings nor the prediction scores but provides insights on the performance of the embeddings in specific subgraphs.

**Evaluation of embeddings.** *ReliK* bears an interesting connection with ranking-based quality measures, in particular with the Mean Reciprocal Rank (MRR) and HITS@K for head, tail, and relation prediction [5, 7, 9, 30, 33, 39]. For a triple $(?, ?, t)$ with unknown head MRR is the average of the reciprocal of ranks of the correct heads in the KG given the relationship $r$ and tail $t$. As such, *ReliK*, can be considered a generalization of MRR as the MRR for triples of the kind $(?, ?, t)$ and $(h, ?, ?)$. Since the triples $(?, r, t)$ are included in $(?, ?, t)$, *ReliK* includes more information than MRR. Moreover, while MRR and HITS@K provide a global indication of performance, *ReliK* is suitable for local analysis. Yet, current global measures have been recently shown to be biased towards high-degree nodes [35].

## 6  CONCLUSION

Aiming to develop a measure that prognosticates the performance of a knowledge graph embedding on a specific subgraph, we introduced *ReliK*, a KGE reliability measure agnostic to the choice of the embeddings, the dataset, and the task. To allow for efficient computation, we proposed a sampling-based approximation, which we show to achieve similar results to the exact *ReliK* at less than half of the time. Our experiments confirm that *ReliK* anticipates the performance on a number of common and complex downstream tasks for KGEs. In particular, apart from correlating with accuracy in prediction and classification tasks, *ReliK* discerns the right answers to complex logical queries and guides the mining of high-confidence rules on subgraphs dense in terms of *ReliK* score. These results suggest that *ReliK* may be used in other domains, as well as a debugging tool for KGEs.

**Ethical use of data.** The measurements performed in this study are all based on datasets that are publicly available for research purposes. We site the original sources.

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

# A   APPENDIX

## A.1   Embeddings

In our experiments, we compare the following embedding methods.

- **TransE** [7] is the first translational model. The score of a triple is obtained by the difference $-\|\mathbf{e}_h + \mathbf{e}_r - \mathbf{e}_t\|_p$ between the head embedding $\mathbf{e}_h$ translated by the relation $\mathbf{e}_r$ and the tail embedding $\mathbf{e}_t$. The score ranges in $[-\infty, 0]$ with positive triples close to 0. This is a strong baseline for all previous works [2, 29, 39].

- **DistMult** [43] is a notable representative of the semantic similarity family. The score $\mathbf{e}_h^\top \mathrm{diag}(\mathbf{W}_r)\mathbf{e}_t$ is bilinear and the relation is a square diagonal matrix $\mathbf{W}_r$. The score ranges in $[-\infty, +\infty]$, whereby positive triples are assigned higher scores.

- **RotatE** [33] is a representative of the complex vector family, whereby vector values are complex numbers. The score $-\|\mathbf{e}_h \circ$ $\mathbf{e}_r - \mathbf{e}_t\|$ is the analogous of TransE's score in the complex space and ranges in $[-\infty, 0]$ with positive triples scoring close to 0.

- **PairRE** [9] is a more recent asymmetric version of TransE in which the relations are represented by two vectors, an head relation $\mathbf{e}_{rh}$ and a tail relation $\mathbf{e}_{rt}$. The score $-\|\mathbf{e}_h \circ \mathbf{e}_{rh} - \mathbf{e}_t \circ \mathbf{e}_{rt}\|$ is an enriched version of TransE and ranges in $[-\infty, 0]$ with positive triples scoring close to 0.

- **ComplEx** [38] uses complex evaluated embeddings. The score function is the real part $Re$ of the complex trilinear dot-product among the embedding of a triple $Re(\langle \mathbf{e}_r, \mathbf{e}_h, \overline{\mathbf{e}_t} \rangle)$.

- **ConvE** [14] applies a multilevel convolutional network with filters $\omega$ on the head and relation embeddings. The resulting tensor is then projected to a vector by a linear layer with parameters $\mathbf{W}$ and multiplied to the tail vector. The scoring function is $f(vec(f([\overline{\mathbf{e}_h}; \overline{\mathbf{e}_r}] * \omega))\mathbf{W})\mathbf{e}_t$.