# OpenReview forum: "ReliK: A Reliability Measure for Knowledge Graph Embeddings"
_ACM.org/TheWebConf/2024/Conference — TheWebConf24_

### Official Review · Reviewer_uWBF · 2023-10-29

**Novelty:** 5
**Technical Quality:** 5

**Review:**

This work proposes Relik, a novel metric to measure the reliability/performance of knowledge graph embeddings on various downstream tasks before actually employing them on those tasks. Experiments with six KGE methods across both simple and advanced downstream tasks demonstrate that Relik scores do correlate with task performance to varying extents.

Overall this is a novel and solid work: I like the idea of pre-determining KGE effectiveness given the costly nature of deploying them, the generality of the proposed approach (across KGE methods and tasks), and more. I have a few comments that hopefully would be helpful to  the authors.

- Six KGE approaches (Table 1) are adopted to test out Relik in this work. While I understand there are at least 100+ KGE approaches and it is impossible to cover every single one, I wonder if the current selection is adequate from a representation standpoint: for example, graph and graph neural network-based approaches seem to be underrepresented, such as CompGCN [1]. Certain methods with strong empirical performance, such as NBFNet [2], might also be an interesting choice. Other works published at the WebConf, such as KRACL [3] and RED-GNN [4], could also be additions. It might be helpful to clarify why the authors believe the current selection of KGE approaches is representative of the research landscape (in order to claim generality), or to include and/or acknowledge additional KGE approaches in this work.

- According to Table 3, Relik seems to work really well with relation prediction tasks, but not so well with classification tasks. I wonder if the authors might have explanations for these task performance disparities.

- It would be nice to provide a qualitative analysis, presenting examples of specific triples and Relik scores, to highlight why a reliability measure (especially a local one) would be valuable in application scenarios.

[1] Vashishth, Shikhar, et al. "Composition-based Multi-Relational Graph Convolutional Networks." International Conference on Learning Representations. 2019.

[2] Zhu, Zhaocheng, et al. "Neural bellman-ford networks: A general graph neural network framework for link prediction." Advances in Neural Information Processing Systems 34 (2021): 29476-29490.

[3] Tan, Zhaoxuan, et al. "KRACL: contrastive learning with graph context modeling for sparse knowledge graph completion." Proceedings of the ACM Web Conference 2023. 2023.

[4] Zhang, Yongqi, and Quanming Yao. "Knowledge graph reasoning with relational digraph." Proceedings of the ACM web conference 2022. 2022.

**Questions:**

please see above

**Reviewer Confidence:**

3: The reviewer is confident but not certain that the evaluation is correct

**Scope:**

3: The work is somewhat relevant to the Web and to the track, and is of narrow interest to a sub-community

---

### Official Review · Reviewer_L92N · 2023-11-14

**Novelty:** 5
**Technical Quality:** 5

**Review:**

Sunmary:

This paper propose a reliable metric Relik for KGE models. Relik is a task- and model-agnostic metric without no further training, which can be used in link prediction / triple classification / rule mining / question answering.

Strengths:

- Knowledge graph embedding evaluation is an important topic for KG research
- The authors provide a sound theoretical analysis to propose such an evaluation metric
- Experiments are more adequate and involve diverse tasks and datasets

Weaknesses:

- A comprehensive comparasion should be conducted among Relik and other newly proposed KGE evaluation metrics [1,2,3]. The starting point and focus proposed by these methods may be different, but they are all aimed at better assessing the quality of the KGE. A detailed comparison is necessary.
- Some important KGE methods are not evaluated in the experiments. For example, gnn-based methods like RGCN, CompGCN. and semantic-based methods TuckER and text-based methods like KG-BERT.



[1]. Do Pre-trained Models Benefit Knowledge Graph Completion? A Reliable Evaluation and a Reasonable Approach
[2]. Rethinking knowledge graph evaluation under the open-world assumption
[3]. Rethinking graph convolutional networks in knowledge graph completion

**Questions:**

I hope the authors can address the issues mentioned in my review. If all my concerns are resolved, I will consider raising my score.

**Reviewer Confidence:**

3: The reviewer is confident but not certain that the evaluation is correct

**Scope:**

3: The work is somewhat relevant to the Web and to the track, and is of narrow interest to a sub-community

---

### Official Review · Reviewer_vCDo · 2023-11-21

**Novelty:** 3
**Technical Quality:** 4

**Review:**

The research paper titled "ReliK: A Reliability Measure for Knowledge Graph Embeddings" addresses the challenge of assessing the performance of knowledge graph embeddings (KGEs) on specific downstream tasks and in particular parts of a knowledge graph (KG). Knowledge graph embeddings represent entities and relationships as vectors, and their effectiveness has been demonstrated in various web-related tasks. However, there is a lack of a unified measure to anticipate how well a given KGE will perform in a specific downstream task and local to a specific individual triple.

The authors introduce "ReliK," a reliability measure for KGEs that relies solely on KGE embedding scores. Unlike existing methods, ReliK is task-agnostic, requiring no additional KGE training.  the authors show that ReliK correlates well with both common downstream tasks (e.g., tail/relation prediction, triple classification) and advanced tasks (e.g., rule mining, question answering) while preserving locality. The provided method is rather a generalization of MRR.

The paper is well-written and easy to understand. The introduced measure Relik measure is clear concise and simple. The measure combines both the head-rank and tail-rank in a straightforward manner, taking the average of their reciprocals. The rationale behind the measure is intuitive  and straight-forward also the approximation is direct. The provided experiments covers several KGE methods and downstream tasks.

**Questions:**

1. Can ReliK replace other measures such as MRR, Acc, etc.?
2. The paper did not provide a computational efficiency comparison between ReliK and the other existing measures. How does ReliK compare to existing reliability measures, particularly in terms of computational efficiency?
3. The paper explicitly asked the question "Is there a measure that provides a prior indication of the performance of a KGE on a specific subgraph?". I would like to ask if there are specific domains or use cases where this reliability measure could have a significant impact. What are these domains that require performance evaluation for specific subgraphs?
4. How does the proposed measure scale with larger knowledge graphs? The experiments include only small-sized graphs except for Yago for rule mining.

Limitations:
1. While the simplicity of ReliK is a positive aspect, it also contributes to the perception of a limited novel contribution. The paper lacks in-depth exploration or extension beyond the simplicity of the proposed measure. This might limit its significance compared to existing measures like MRR and accuracy.

2. The paper lacks a thorough computational efficiency comparison between ReliK and existing reliability measures.

3.  The paper mentions the question of whether there is a measure that provides a prior indication of the performance of a KGE on a specific subgraph. However, the discussion on specific domains or use cases where ReliK could have a significant impact is somewhat limited.

**Ethics Review Description:**

-

**Reviewer Confidence:**

3: The reviewer is confident but not certain that the evaluation is correct

**Scope:**

3: The work is somewhat relevant to the Web and to the track, and is of narrow interest to a sub-community

---

### Official Review · Reviewer_7ES7 · 2023-11-28

**Novelty:** 5
**Technical Quality:** 5

**Review:**

In this article, the authors present ReliK, a reliablilty measure aiming at determining the quality of Knowledge Graph Embeddings (KGEs) while being local, agnostic to some specific KGEs, the task, and the KGs, and without needing re-training KGEs.
To do so, the author provide a detailed study that discusses the requirements for such a metric, their proposal, its complexity, lower and upper bounds to ease computation, and evaluation on 4 tasks, with 6 KGEs, on 6 datasets.
They show that the upper bound is a great proxy for ReliK, and that it adequately correlates on usual metrics (MRR) for the task of link prediction and triple classification, and is also able to provide improvements for the tasks of rule mining and query answering.

Overall, I appreciate the work and the effort put by the authors in providing a detailed and clear description of their metric, the motivation, and the lower/upper bounds and complexity. I think this metric is an advancement in the field of KGEs, especially (as mentioned in their related work) w.r.t. other orthogonal directions such as embedding calibration. The article is also easy to read and follow.

I have some major and minor comments as follows:

Major comments:

(MC1) You claim that this metric can be used to anticipate the performance of a given KGE in downstream tasks and you experiment on link prediction, triple classification, rule mining and query answering. However, for query answering and rule mining, you only show that ReliK values are higher for true answers/rules than for false ones. You do not provide a correlation study similar to the one for link prediction and triple classification. This puzzles me since it does not allow to qualify whether ReliK is good at anticipating the performance of KGEs on these specific tasks. It only shows that it can be used to address the task. I expected the same kind of study as for link prediction and triple classification.

(MC2) Your claim that your metric is task agnostic is strong. However, you do not evaluate on a broader variety of tasks including e.g. entity classification or clustering. Could your metric be applicable to such tasks? Otherwise could you specific the task-specific requirements to narrow down the set of considered tasks?

(MC3) You mention that you are agnostic to KGEs. However, it seems you are relying on some assumptions:
--> KGEs are expected to output higher scores for correct triples than for incorrect ones and not the other way around which could theoretically be possible for a KGE working in [0, +inf] with 0 being for correct triples and +inf being for incorrect ones. In such a case, we would just need to use -scoreKGE for your metric to be applicable but this is something to take into account.
--> You only test with KGEs designed for link prediction. Could other KGE models like RDF2Vec for data mining be applicable? This may be related with MC2.  Otherwise, could you specify the requirements for KGEs on which your metric would be applicable?

Minor comments:
(mc1) "Computing ReliK takes $\mathcal{O}(|\mathcal{E}| \cdot |\mathcal{R}|)$: why is there $|\mathcal{R}|$ in Equation (1)? It seems to only require to test negative triples by changing the head and the tail of the triple.

(mc2) Using $rank^S_H$ in Equations (3), (4), (5), (6) to count the number of incorrect triples that have a score greater than the ground truth leads to forgetting the +1 factor in the definition of $rank^S_H$. Hence, a KGE ranking the ground truth last would lead, e.g., to have $\frac{1}{|\mathcal{N}^{-}(h)| + 1}$ in Equations (5) and (6). It does not change the global results in my opinion that is why this is a minor comment. But I think it thus does not *exactly* corresponds to the description, even though the intuition remains the same.

(mc3) You mention that the performance of KGEs for several applications varies across the parts of the KG. While I do not question this possibility, would you have a reference to sustain the claim?

Typos:
- sampling step in Line 5 -> Line 1?
- "person score" -> Pearson score?
- "with unknown head[,], MRR is the average"

Comments for future work:
You mentioned that ReliK can help evaluate how well embeddings capture the semantics included in a KG. I thus wonder how it could relate to
- Inc@K: https://link.springer.com/chapter/10.1007/978-3-030-88361-4_24
- Sem@K: https://www.semantic-web-journal.net/content/semk-my-knowledge-graph-embedding-model-semantic-aware-0


Edit post rebuttal:
I would like to thank the authors for their detailed answers to my comment. mc1, mc2, and mc3 have been fully addressed, as well as MC1. I still think it would be great to further demonstrate the generality of the metric (MC2, MC3) by testing on other tasks that do not involve triple ranking (e.g. node classification) with embedding models that were not design to rank triples (e.g., RGCN, or RDF2Vec). However, I think the metric is of interest to the community and the paper is well-written.

**Questions:**

Could you comment mainly on MC1, MC2, MC3, and mc1?
I would also appreciate any comments you may have on mc2, and mc3.

**Ethics Review Description:**

-

**Reviewer Confidence:**

3: The reviewer is confident but not certain that the evaluation is correct

**Scope:**

3: The work is somewhat relevant to the Web and to the track, and is of narrow interest to a sub-community

---

### Official Review · Reviewer_1yJK · 2023-11-29

**Novelty:** 4
**Technical Quality:** 6

**Review:**

The authors propose a new measure for evaluating knowledge graph embedding approaches.
They show that the measure correlates with head and tail prediction as well as classification tasks.

Overall, the paper is well-written and easy to follow.
In some parts, more details would make it easier to understand e.g. the measure N^{-} is critical for the approach but should be explained in more detail.
It really depends which triples are assumed to be negative. If all relations and entities are used,
the space is quite huge and probably also contains at least some true positives. In figure 2, only a few specific relations are used.

Another crucial point in the motivation is that the ReliK measure should be a proxy for all other downstream tasks.
But this measure is again something that could be optimized for and could also be seen as its own task.
The main question is why exactly this task should be a proxy for all other (downstream) tasks.

There are many other tasks that can be solved by KGE that have nothing to do with head/tail prediction
e.g. (1) predicting if an entity is a person or not, (2) if a movie has a high rating, (3) semantic analogies etc.
The ReliK measure is mainly also based on how well a model can perform the head/tail prediction and thus might be a good proxy for those tasks
but there are many more tasks as well.

The evaluation is done very well (in terms of analyses etc).
I would also love to see if the approximations work as well on other datasets (the runtime of ReliK on Countries dataset is between 5-10 seconds;
thus I assume it could also be exactly computed for e.g. Codex-S, which of course has more entities and relations but should still be feasible even if it takes some hours/days).
It would also show that the approximations are performing as well as ReliK on more complex datasets as well.

In the main evaluation (Table 3), it is shown that for some KGE approaches and the tail prediction task
the ReliK measure is not correlating (e.g. DistMult on Codex-S; PairRE on Codex-M and Codex-L etc).
If it fails on those cases why is it a good metric in general? Also, in the classification task PairRE, ComplEx, and ConvE are not correlating with the proposed measure, or am I missinterpreting the table?

The locality property of the measure is very interesting. It can show that in some subgraphs, the KGE is not doing well.
It would be interesting to see some of the cases and also find out why the KGE is performing poorly on this subset.
On the other hand, you want most of the time, a general measure of how good the KGE is for a specific downstream task for the whole graph and not only for a part of it.
But I think this is a good way to "debug" KGE approaches and see where they are performing well/poorly.

Overall, the paper should also mention the limitations of the approach. Example:
As far as I understood, ReliK is specific for geometric models which are able to provide a score for a given triple to be contained in the KG or not.
Thus, the approach can not easily be applied to models such as Node2vec, OWL2Vec or any graph neural network based approach like GraphSAGE [1], Graph ATtention Network (GAT) [2] etc.

Another relevant work on evaluating KGE is
Pellegrino, M.A., Cochez, M., Garofalo, M., Ristoski, P. (2019). A Configurable Evaluation Framework for Node Embedding Techniques.
In: Hitzler, P., et al. The Semantic Web: ESWC 2019 Satellite Events. ESWC 2019.

In addition, it would be good to have a reference implementation hosted somewhere to be able to reproduce the results and apply the measure to other KGs.

Small hint: Section 4 Datasets: The initial version of YAGO is extracted from Wikipedia - either update the reference or change the description.

[1] Inductive Representation Learning on Large Graphs. W.L. Hamilton, R. Ying, and J. Leskovec. Neural Information Processing Systems (NIPS), 2017.
[2] Graph Attention Networks. P. Veličković et al. International Conference on Learning Representations (ICLR) 2018


Pros:
- the authors are analyzing the results in various ways (e.g. subgraph size, etc)
- an approximation measure which can be different for various subgraphs (and thus show where the embedding might be good or bad)
- good possibility for "debugging" KGE approaches (analyze why a KGE approach is bad on certain subgraphs)

Cons:
- why is the proposed task/ metric representative for all?/ many downstream tasks?
- approximations of ReliK measure only tested empirically on a very small and simple dataset (Countries)
- also mention the limitations of the approach e.g. can not be easily used for Node2vec, GraphSAGE etc
- provide a clear definition of N^{-}
- provide a reference implementation of the measures

**Questions:**

- Page 7 right column: "We evaluate ReliK on the ability to detect whether an instance of an answer is true or false."
Do you mean if an entity is in the result set of a query or not? If that is not the case, elaborate a bit more on this.
- How does ReliK indicate that the embedding needs further tuning? (page 6)
- in the classification task PairRE, ComplEx, and ConvE are not correlating with the proposed measure, right?

**Reviewer Confidence:**

2: The reviewer is willing to defend the evaluation, but it is likely that the reviewer did not understand parts of the paper

**Scope:**

4: The work is relevant to the Web and to the track, and is of broad interest to the community

---

### Decision · Program_Chairs · 2024-01-22

**Decision:**

Accept

**Comment:**

This paper introduces ReliK, a reliability measure designed to assess the quality of Knowledge Graph Embeddings (KGEs). This measure is unique in its local applicability, agnostic nature regarding specific KGEs, tasks, and KGs, eliminating the need for KGE re-training. The study delves into the metric's prerequisites, complexity, and computation bounds, offering a comprehensive evaluation. It is evident that the upper bound effectively mirrors ReliK's performance, showcasing strong correlation with standard metrics (MRR) in link prediction and triple classification. Additionally, ReliK demonstrates enhancements in rule mining and query answering tasks.

 The paper fits well to the Semantics track. Overall the reviewers consider this work as novel, solid, and sufficiently technically sound.
 The authors provide additional evaluation results to address the criticism for limited generalizability and missing experiments, which has been acknowledged by the reviewers.

 Pros:
 1. The paper is well-written and easy to follow.
 2. The introduced metric represents a notable advancement in KGEs.
 3. The authors offer a robust theoretical analysis to introduce this evaluation metric.
 4. The approximation measure varies across subgraphs, revealing areas of strong or weak embedding performance.
 5. The proposed method presents a valuable opportunity for diagnosing issues in KGE approaches.
 6. The experiments provided are comprehensive, encompassing various KGE methods and downstream tasks.

 Cons:
 1. The absence of a reference implementation for the proposed measures has been criticized.
 2. While ReliK's simplicity is a positive aspect, it also limits the perceived novelty of the contribution, lacking an in-depth exploration beyond its straightforward measure.
 3. The paper doesn't thoroughly compare the computational efficiency between ReliK and existing reliability measures.
 4. The discussion about specific domains or impactful use cases for ReliK seems limited.
 5. It is not completely why the proposed metric is considered representative for many downstream tasks. A comprehensive comparison with other recently proposed KGE evaluation metrics should be conducted.